# Comparative Study of IGBT and SiC MOSFET Three-Phase Inverter: Impact of Parasitic Capacitance on the Output Voltage Distortion

**Paisak Poolphaka, Ehsan Jamshidpour** [ID]**, Thierry Lubin ***[ID]**, Lotfi Baghli** [ID] **and Noureddine Takorabet** [ID]

Groupe de Recherche en Electrotechnique et Electronique de Nancy (GREEN), Université de Lorraine, 54506 Vandoeuvre-lès-Nancy, France; paisak.poolphaka@univ-lorraine.fr (P.P.); ehsan.jamshidpour@univ-lorraine.fr (E.J.); lotfi.baghli@univ-lorraine.fr (L.B.); noureddine.takorabet@univ-lorraine.fr (N.T.)
* Correspondence: thierry.lubin@univ-lorraine.fr; Tel.: +33-(0)372745095

**Abstract:** This study investigates the nonlinearities in three-phase inverters for SiC-based systems and compares their performance to IGBT-based systems. An analytical model of inverter voltage distortion is developed, which accounts not only for dead time (td), switching delay time, switching frequency (fs), and voltage drops of power devices, but also for output parasitic capacitance (Cout). Experimental tests validate the model, which provides a more accurate estimate of the inverter's output phase voltage distortion. The power device characteristics are obtained from datasheets, while Cout is determined through experimentation. Three-phase inverters with varying switching frequencies, fundamental frequencies, and dead-time values are used in simulations and experiments to determine the influence of nonlinearity on phase voltage deviation and current distortion. The results show that, due to SiC devices' faster switching time, the phase voltage deviation and phase current distortion are lower in SiC-based inverters than in IGBT-based ones for high-frequency applications, as the dead time can be reduced.

**Keywords:** DC/AC converter; three-phase inverter; high frequency applications; performance comparison





## 1. Introduction

In pursuit of reducing reliance on fossil fuels and transitioning to renewable energy sources, the electrification of numerous industrial systems stands as a primary objective for many countries. Moreover, in efforts to mitigate greenhouse gas emissions and air pollution, electrifying transportation systems can play a crucial role [1]. Consequently, the utilization of various types of electrical machines, such as synchronous machines, has witnessed significant growth in recent years [2,3].

High-speed electric motor drive systems have gained significant popularity in various industries due to their compact size, lightweight design, and impressive power density. These systems operate at exceptionally high fundamental frequencies ($f_0$), typically in the kilohertz range, necessitating inverters capable of handling such demanding switching frequencies.

However, traditional silicon (Si) IGBT technology has limitations in achieving high switching frequencies, resulting in compromised current quality at higher fundamental frequencies [4,5]. The decrease in the frequency modulation index ($f_s/f_0$) with increasing fundamental frequency leads to higher total current harmonic distortion (THD) [6]. Additionally, Si-IGBT technology exhibits drawbacks in terms of efficiency, operating temperature, and power density.

To overcome these limitations, SiC MOSFET technology has emerged as a superior alternative to Si-IGBT technology. SiC MOSFETs offer advantages such as higher

efficiency [7–9], increased switching frequency, extended operating temperature capabilities [10,11], and improved power density [12]. These properties make SiC MOSFETs a preferred choice for high-speed electric motor drive systems, surpassing the limitations of Si-IGBT technology.

In this context, SiC-based inverters have been identified as suitable for high-speed machines requiring high fundamental and switching frequencies to minimize total harmonic distortion of the current [13]. The voltage deviation and current distortion in the inverter are directly proportional to the increasing $f_s$.

An expanded version of our previous conference paper [14] is presented in this paper, with the primary objective of providing a comprehensive model and investigation of the varied factors affecting voltage and current distortion in SiC-based and IGBT-based inverters designed for high-speed machinery applications [15]. These factors, including dead time, switching delay time, voltage drop, and the parasitic capacitance of the components, are the focus of the study.

A wide range of fundamental frequencies, reaching up to 3400 Hz, and switching frequencies ranging from 10 kHz to 100 kHz are covered in our experimental analysis. The operational limits of IGBT inverters when employed in high-speed motor drives, a prevalent choice in applications such as electric vehicles, airplanes, electric ships, and more, are sought to be anticipated. Furthermore, the advantages of employing SiC inverters will be elucidated.

The significance of considering the parasitic capacitance of both IGBT and SiC systems is underscored to ensure precise predictions regarding inverter frequency limits. The experimental validation of these limitations, extending up to a PWM frequency of 100 kHz, represents a significant advancement in the field, building upon prior research contributions [12,16].

The remainder of this paper is organized as follows. Section 2 presents the proposed analytical model. Section 3 explores the influence of imperfections in the considered components in detail. Section 4 presents the experimental results. Finally, the conclusion is provided in the last section.

## 2. Analytical Modeling of a Three-Phase Voltage Source Inverter and Imperfections

### 2.1. Three-Phase Voltage Source Inverter Modeling

Figure 1 illustrates a three-phase voltage source inverter (3Ph-VSI) used for general applications. In this figure, the load current is positive when it flows from the source to the load, and negative when it reverses its direction.

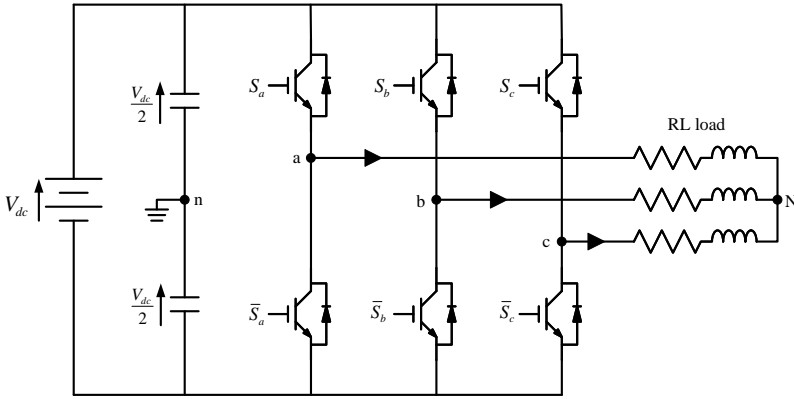

**Figure 1.** Three-phase Voltage Source Inverter.

The modeling of the 3Ph-VSI is initially developed without considering imperfections such as dead times, delays in component startup and shutdown, voltage drops, and parasitic capacitances. Therefore, the ideal switching function can be written as a simple arithmetic relationship between the control signals of the switches defined in Figure 1,

such that $S_i + \bar{S}_i = 1$ (where $i : a, b,$ or $c$). Consequently, the voltages $V_{an}, V_{bn}, V_{cn}$ can be expressed as follows:

$$\begin{cases} V_{an} = (2S_a - 1)\frac{V_{dc}}{2} \\ V_{bn} = (2S_b - 1)\frac{V_{dc}}{2} \\ V_{cn} = (2S_c - 1)\frac{V_{dc}}{2} \end{cases} \tag{1}$$

Assuming that the load is connected in a star configuration and is balanced, the three-phase output voltages $V_{aN}, V_{bN},$ and $V_{cN}$ can be expressed using the matrix relationship as follows (2):

$$\begin{bmatrix} V_{aN} \\ V_{bN} \\ V_{cN} \end{bmatrix} = \frac{V_{dc}}{3} \begin{bmatrix} 2 & -1 & -1 \\ -1 & 2 & -1 \\ -1 & -1 & 2 \end{bmatrix} \begin{bmatrix} S_a \\ S_b \\ S_c \end{bmatrix} \tag{2}$$

### 2.2. Analytical Modeling of Imperfections

In this section, an analytical model will be developed to consider various imperfections in a 3Ph_VSI. Factors such as dead times between switching signals, switching times, voltage drops caused by power switches (IGBT, SiC, diode), and the effects of parasitic capacitances will be taken into account. The focus of the study is on analyzing one leg of the VSI, as illustrated in Figure 2.

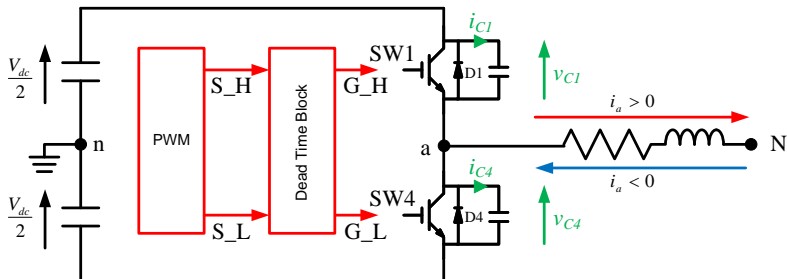

**Figure 2.** A-phase leg of three-phase inverter with the dead-time block.

In the following, the distortion voltage due to imperfections is defined as $v_{an}^{err}$, which is the difference between the actual output voltage ($v_{an}$) and the ideal output voltage ($v_{an}^*$), corresponding to a perfect inverter:

$$v_{an}^{err} = v_{an}^{err1} + v_{an}^{err2} + v_{an}^{err3} + v_{an}^{err4} = v_{an} - v_{an}^* \tag{3}$$

where

- $v_{an}^{err1}$ is the average voltage drop over one modulation period (PWM) due to dead times;
- $v_{an}^{err2}$ is the average voltage drop due to component switching times;
- $v_{an}^{err3}$ is the average voltage drop due to voltage drops across power switches;
- $v_{an}^{err4}$ is the average voltage drop caused by the effects of parasitic capacitances.

### 2.2.1. Dead Time Effect

Ideally, based on Figure 2, the power switches S1 and S4 are controlled by PWM signals ($S_H$, $S_L$). In reality, a dead time ($t_d$) needs to be introduced where both S1 and S4 turn off at the same time to prevent a short circuit from occurring through S1 and S4. During the dead-time interval, the phase current flows through diode D4, resulting in an output voltage equal to $-\frac{V_{dc}}{2}$ when the current is positive ($i_a > 0$). When the current is negative ($i_a < 0$), the phase current flows through diode D1, causing the output voltage to

become positive $\frac{V_{dc}}{2}$. The average voltage drop caused by the dead time (the yellow area in Figure 3) can be expressed as follows:

$$v_{an}^{err1} = \Delta V_1 sign(i_a) \qquad with \ \Delta V_1 = -V_{dc}\left(\frac{t_d}{T_s}\right) \tag{4}$$

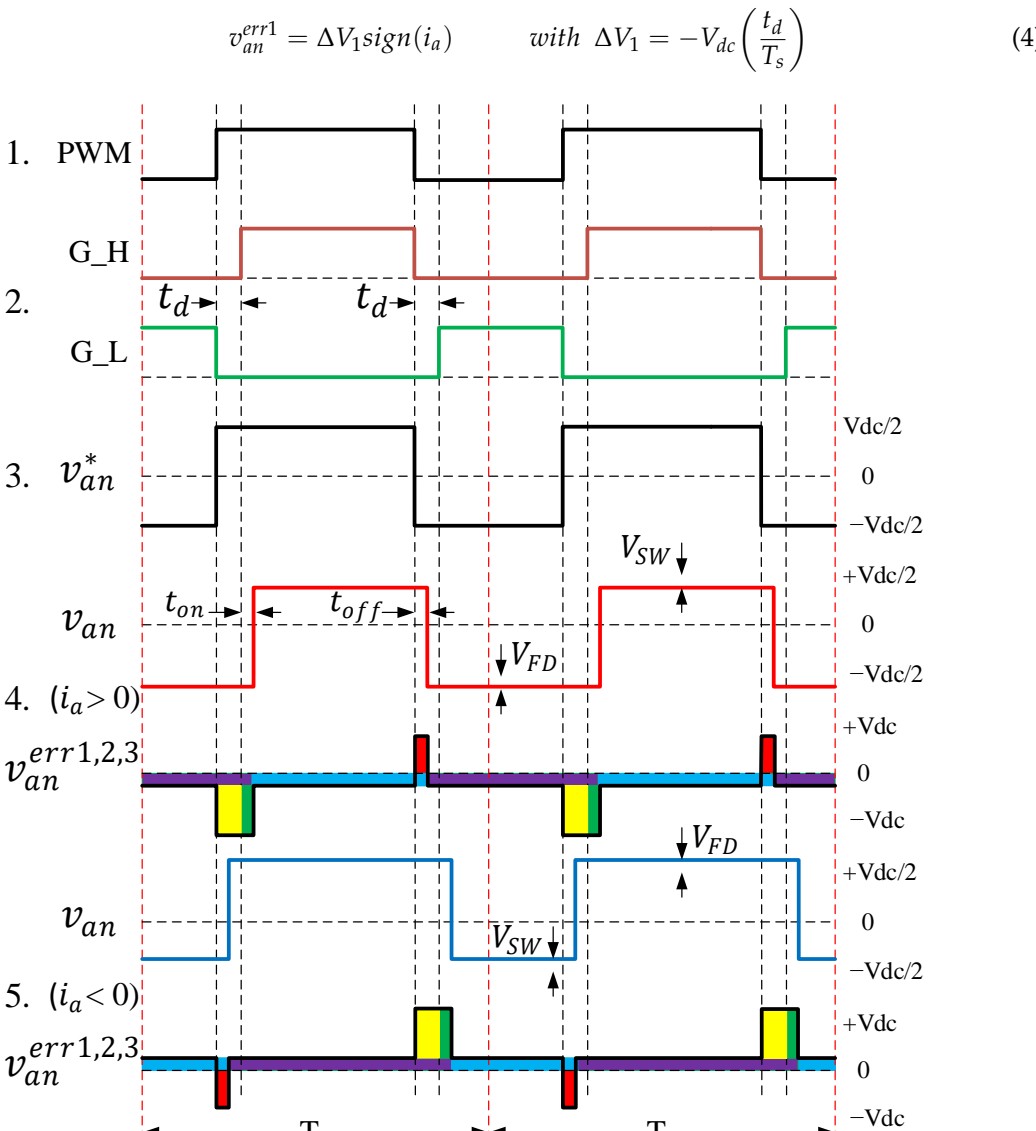

**Figure 3.** Waveform of the output voltage taking into account voltage drops, dead time, switching times, and parasitic capacitors.

The relation (4) demonstrates that the average voltage drop induced by the dead time is proportional to the ratio $\frac{t_d}{T_s}$. When the PWM frequency is increased ($T_s$ is decreased), the voltage drop becomes more significant if the dead time cannot be reduced in the same proportion due to limitations imposed by the drivers or the power switches (IGBT or SiC). The average distortion voltage at the terminals of phase "a" induced by the dead time can be expressed, based on Equation (4), according to the directions of the three phase currents, as follows:

$$v_{aN}^{err1} = \frac{\Delta V_1}{3}\begin{bmatrix} 2 & -1 & -1 \end{bmatrix}\begin{bmatrix} sign(i_a) \\ sign(i_b) \\ sign(i_c) \end{bmatrix} \tag{5}$$

### 2.2.2. Rising Time and Falling Time Effect

The switching of power electronic components is not instantaneous. In practice, it takes a certain amount of time for a switch to turn on and a certain amount of time to ensure its blocking. These switching times accumulate with the dead time defined in the previous section and must be considered to accurately determine the waveform of the voltage $V_{an}$. Therefore, when $i_a > 0$, the average voltage drop induced by the switching times $t_{on}$ (green area in Figure 3) and $t_{off}$ (red area in Figure 3) can be expressed as follows:

$$v_{an}^{err2} = \Delta V_2 sign(i_a) \qquad with \ \Delta V_2 = -V_{dc}\left(\frac{t_{on} - t_{off}}{T_s}\right) \tag{6}$$

where:

$t_{on}$ is the turn-on time ($t_{on_d} + t_r$), with $t_{on_d}$ being the propagation time at turn-on, and $t_r$ being the rising time (from $10\%I_c$ to $90\%I_c$).

$t_{off}$ is the turn-off time ($t_{off_d} + t_f$), with $t_{off_d}$ being the propagation time at turn-off, and $t_f$ being the fall time (from $90\%I_c$ to $10\%I_c$).

From Equation (6), the voltage drop across phase "a" induced by the switching times can be expressed based on the directions of the three phase currents as follows:

$$v_{aN}^{err2} = \frac{\Delta V_2}{3}\begin{bmatrix} 2 & -1 & -1 \end{bmatrix}\begin{bmatrix} sign(i_a) \\ sign(i_b) \\ sign(i_c) \end{bmatrix} \tag{7}$$

### 2.2.3. Consideration of Voltage Drops in Components

In power electronic systems, voltage drops across components such as power switches (IGBT, SiC) and diodes should be taken into account. These voltage drops depend on the direction of the phase current. Let $V_{SW}$ represent the voltage drop across a conducting switch, identified by the blue area in Figure 3. With this in mind, the following relationship can be formulated:

$$v_{an} = \frac{V_{dc}}{2} - V_{SW} \tag{8}$$

with $V_{SW} = V_{sw0} + R_{sw_{on}}|I_a|$, which represents the voltage drop across the conducting switch, where $V_{sw0}$ is the forward voltage drop of the switch and $R_{sw_{on}}$ is the on-state resistance of the switch. When calculating the voltage drop, the current $|I_a|$ is assumed to remain constant over a PWM period. As illustrated in Figure 3 when $i_a > 0$ and $G_H$ is in a low state while $G_L$ is in a conducting state, the current then flows through diode D4. During this conduction phase, the output voltage $v_{an}$ takes the following value:

$$v_{an} = \frac{V_{dc}}{2} - V_{FD} \tag{9}$$

with $V_{FD} = V_{F0} + r_f|I_a|$, which represents the voltage drop across the conducting diode, where $V_{F0}$ is the forward voltage drop of the diode and $r_f$ is the on-state resistance of the diode. When calculating the voltage drop, the current $|I_a|$ is assumed to remain constant over a PWM period. By combining both terms, the average output voltage drop resulting from the voltage drops in the components can be rewritten as a function of the sign of the current in the corresponding phase (in this case, phase a).

$$v_{an}^{err3} = \Delta V_3 \ sign(i_a) \qquad with \ \Delta V_3 = -(V_{SW}.D + V_{FD}.(1 - D)) \tag{10}$$

The average voltage drop across phase "a" induced by the voltage drops in the components can be expressed in terms of the directions of the three-phase currents as follows:

$$v_{aN}^{err3} = \frac{\Delta V_3}{3} \begin{bmatrix} 2 & -1 & -1 \end{bmatrix} \begin{bmatrix} sign(i_a) \\ sign(i_b) \\ sign(i_c) \end{bmatrix} \tag{11}$$

### 2.3. Effects of Parasitic Capacitance

Power semiconductor models (IGBT, SiC) exhibit parasitic capacitances, particularly the output capacitance between the emitter and collector for IGBTs, and between the drain and source for SiC, as shown in Figure 2. These capacitances, with values around a few nF, influence switching and affect output voltage distortion during dead times, depending on the switch type (IGBT or SiC), and also include wiring parasitic capacitances.

In Figure 4, when the phase current flows in the positive direction, during the on period (S1 is ON), the phase current flows through S1. Then, the voltage across the inverter becomes $V_{an} = \frac{V_{dc}}{2}$ and the bottom parasitic capacitance ($C_4$) is charged. During the dead time interval, the top parasitic capacitance ($C_1$) starts charging and $C_4$ discharges to let the phase current flow. Thus, the voltage across the inverter becomes $V_{an} = -\frac{V_{dc}}{2} + V_{C4}$. Then, $C_4$ discharges and the voltage of $C_4$ ($V_{C_4}$) reaches zero ($V_{an} = -\frac{V_{dc}}{2}$). At this point, the phase current ($i_a$) can be expressed as the sum of the charge current ($i_{C_1}$) and the discharge current ($i_{C_4}$) assuming that $C_1$, $C_4$ are equal ($C_1 = C_4 = C_{out}$), where $C_{out}$ is the total parasitic capacitance, which depends on both the transistor parasitic capacitance and the external capacitance due to the cable's connection. The charge or discharge time ($t_{cd}$) of the parasitic capacitor is represented by:

$$t_{cd} = \frac{2C_{out}(V_{dc} - V_{sw} + V_{FD})}{|i_a|} \tag{12}$$

The current for charging (or discharging) the capacitance in the actual dead time interval ($t_d + t_{on} - t_{off}$) of Figure 4 is called the threshold current ($I_{th}$) and is given by:

$$I_{th} = \frac{2C_{out}(V_{dc} - V_{sw} + V_{FD})}{t_d + t_{on} - t_{off}} \tag{13}$$

Three situations where the absolute value of the phase current ($i_a$) is more than $I_{th}$ ($|i_a| > I_{th}$, brown line), equal to $I_{th}$ ($|i_a| = I_{th}$, purple line) and less than $I_{th}$ ($|i_a| < I_{th}$, red line), are illustrated by Figure 4.

The orange areas of Figure 3 show the voltage drop caused by the parasitic capacitors, then considered as expressions obtained as a function of the phase current [16–19].

When $i_a > 0$ :

1.  if $|i_a| < I_{th}$ ;

$$\Delta v_{an}^{err4} = \frac{t_{dc}}{T_S} \left[ (V_{dc} - V_{sw} + V_{FD}) - \frac{|i_a| t_{cd}}{4 C_{out}} \right] \tag{14}$$

2.  if $|i_a| = I_{th}$ ;

$$\Delta v_{an}^{err4} = \frac{t_{cd}(V_{dc} - V_{SW} + V_{FD})}{2 T_s} \tag{15}$$

3.  if $|i_a| > I_{th}$ ;

$$\Delta v_{an}^{err4} = \frac{C_{out}(V_{dc} - V_{SW} + V_{FD})^2}{T_s |i_a|} \tag{16}$$

The average voltage drop due to parasitic capacitances, $V_{an}^{err4}$, can be rewritten as a function of the sign of the current in the corresponding phase (in this case, phase a).

$$v_{an}^{err4} = \Delta V_4 \, sign(i_a) \tag{17}$$

with

$$\Delta V_4 = \begin{cases} \frac{t_{cd}}{T_s}\left( (V_{dc} - V_{sw} + V_{FD}) - \frac{|i_a|t_{cd}}{4C_{out}} \right), & |i_a| < I_{th} \\ \frac{C_{out}}{T_s|i_a|}(V_{dc} - V_{sw} + V_{FD})^2, & |i_a| \geq I_{th} \end{cases} \tag{18}$$

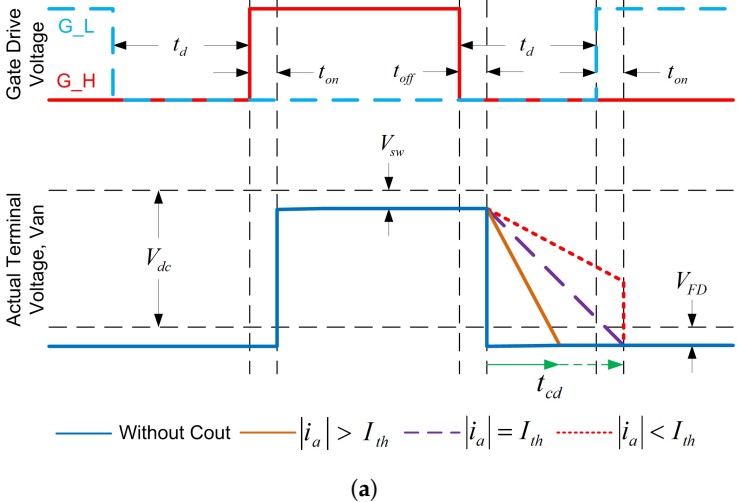

(**a**)

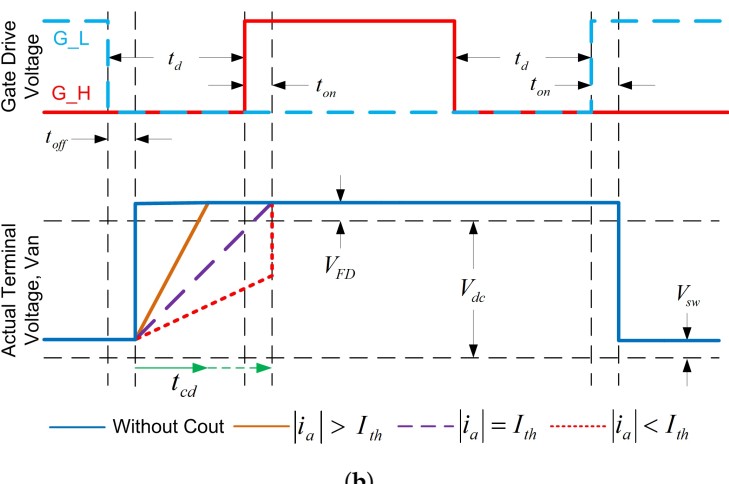

(**b**)

**Figure 4.** Effects of Parasitic Capacitance, (**a**) Switch turn on (**b**) Switch turn off.

In a general sense, the amplitude of the threshold current ($I_{th}$) is much lower than the phase current magnitude $|i_a|$. As a result, the relationship (18) can be simplified as follows:

$$\Delta V_4 = \frac{C_{out}}{T_s|i_a|}(V_{dc} - V_{sw} + V_{FD})^2, \quad i_a \neq 0 \tag{19}$$

The variations of $v_{an}^{err4}$ with respect to the values taken by the phase current $i_a$ are then represented in green in Figure 5 (curve in $\frac{1}{|i_a|}$ for $|i_a| \geq I_{th}$). The voltage drop caused by other imperfections, $V_{an}^{err1,2,3}$ (dead times, switching times, and voltage drop in components), is also depicted in red on this figure, along with the resulting voltage drop shown in blue. It can be observed that the parasitic capacitances have a beneficial effect on the voltage drop, leading to lower values, but this depends on the magnitude of the load current.

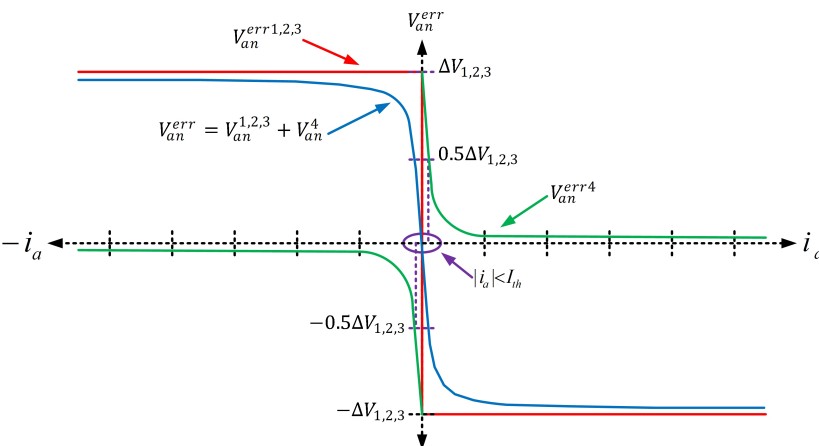

**Figure 5.** The average voltage drop due to parasitic capacitances as a function of $i_a$ is shown in the green curve, and the overall voltage drop (global voltage drop) is represented by the blue curve.

Figure 6 shows the impact of the parasitic capacitance value on the overall voltage drop (limited to $i_a > 0$). A larger parasitic capacitance limits the voltage drop for low currents, but it has little effect on the average global voltage drop at the inverter output for higher currents.

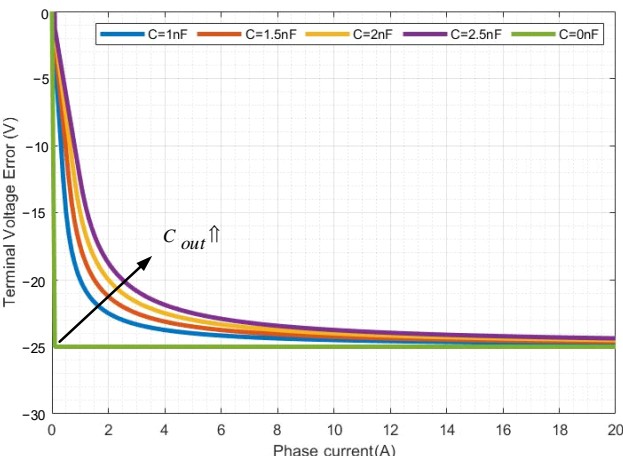

**Figure 6.** Effect of parasitic capacitance value on the output voltage drop.

The average voltage drops resulting from different imperfections of the inverter have been derived in this section. Consequently, the total voltage drop across phase "a" can be expressed as follows:

$$v_{aN}^{err4} = \frac{\Delta V}{3} \begin{bmatrix} 2 & -1 & -1 \end{bmatrix} \begin{bmatrix} sign(i_a) \\ sign(i_b) \\ sign(i_c) \end{bmatrix} \tag{20}$$

where

$$\Delta V = \Delta V_1 + \Delta V_2 + \Delta V_3 + \Delta V_4 \tag{21}$$

Finally, based on Equation (2), this voltage drop can be expressed for all three phases, depending on the signs of the respective currents.

$$\begin{bmatrix} v_{aN}^{err} \\ v_{bN}^{err} \\ v_{cN}^{err} \end{bmatrix} = \frac{\Delta V}{3} \begin{bmatrix} 2 & -1 & -1 \\ -1 & 2 & -1 \\ -1 & -1 & 2 \end{bmatrix} \begin{bmatrix} sign(i_a) \\ sign(i_b) \\ sign(i_c) \end{bmatrix} \tag{22}$$

### 3. The Influence of Imperfections on the Output Voltage of the Inverter

The previous section discussed the average voltage drops resulting from imperfections in the inverter. These imperfections will have two key effects. Firstly, there will be a voltage drop in the RMS value of the fundamental component of the inverter's output voltage. Secondly, the imperfections will generate additional voltage harmonics that will affect the load current, especially the low-frequency harmonics. These effects will be further examined in the following section.

*3.1. RMS Value of the Fundamental Component of the Output Voltage*

The RMS value of the fundamental component at the terminals of a phase ($V_{aN1}$) can be derived by considering the modulation index ($m_a$) and the DC bus voltage ($V_{dc}$):

$$V_{aN1} = \frac{V_{dc}}{2\sqrt{2}} m_a \tag{23}$$

In the case of an inductive load with a phase lag of $\phi_1$, the RMS value of the distortion in the phase voltage $V_{aN}^{err}(t)$ can be expressed as follows:

$$V_{aN1}^{err} = \frac{4\Delta V}{\pi\sqrt{2}} \tag{24}$$

where the term $\Delta V$ has been defined in Equation (21) and encompasses all the voltage drops resulting from the imperfections of the inverter. The value of the voltage drop $V_{aN1}^{err}$ can be evaluated based on the measurement of $V_{a1}$.

$$V_{aN1}^{err} = \frac{-2V_{a1}\cos\phi \pm \sqrt{(2V_{a1}\cos\phi)^2 + 4(V_{a_{\text{ref}}}^2 - V_{a1}^2)}}{2} \tag{25}$$

The voltage $V_{a_{ref}}(t)$ represents the fundamental component of the voltage across phase "a" that would occur in the absence of any imperfections.

*3.2. Harmonics Generated by the Inverter's Imperfections*

The literature focuses on the impact of dead times on harmonic amplitudes, leading to complex formulas [20–22]. Dead times increase harmonics and decrease the fundamental. THD of output voltage increases linearly with dead time. Dead times also generate low-frequency harmonics, which are harder to filter and harmful for synchronous machines [23]. The presence of these low-frequency harmonics is explained based on the voltage $V_{aN}^{err}(t)$ and its spectral decomposition is given by:

$$V_{aN}^{err}(t) = \frac{4}{\pi}\Delta V \sum_{n=1}^{\infty} \frac{1}{n} \sin(n\omega_1 t) \tag{26}$$

where $\Delta V$ is defined in Equation (21), $n$ represents the harmonic order ($n = 5, 7, 11, 13, \ldots$), and $\omega_1$ is the fundamental frequency. The following relationship is obtained through the development of calculations:

$$\begin{aligned} v_{aN}(t) &= \frac{V_{dc}}{2m_a} \sin(\omega_1 t + \phi) + V_{aN}^{err}(t) \\ &= \left[ \frac{V_{dc}}{2m_a} \sin(\omega_1 t + \phi) + \frac{4}{\pi}\Delta V \sin(\omega_1 t) \right] + \frac{4}{\pi}\Delta V \left[ \frac{1}{5}\sin(5\omega_1 t) + \frac{1}{7}\sin(7\omega_1 t) + \ldots \right] \end{aligned} \tag{27}$$

Low-frequency harmonics will be introduced into the voltage across the load due to inverter imperfections. In the following section, these low-frequency harmonics will be experimentally demonstrated.

## 4. Experimental Results

To confirm the validity of our study, experimental tests were conducted on three different switches, including two IGBTs and one SiC MOSFET. The parameters of the switches are provided in Table 1. Table 2 presents the specification of the 3Ph-VSI.

**Table 1.** Component parameters.

| Parameters | IGBT1 (SEMiX251GD126HD) | IGBT2 (SKM100GB125DN) | SiC MOSFET (CCS050M12CM) |
|---|---|---|---|
| $V_{CE}$ or $V_{DS}$ (V) | 1200 | 1200 | 1200 |
| $I_C$ or $I_D$ (A) at 25 ºC | 242 | 100 | 87 |
| $I_F$ (A) at 25 ºC | 207 | 95 | 102 |
| $r_{CE}$ or $r_{DS(on)}$ (mΩ) | 7 | 22.5 | 25 |
| $r_F$ (mΩ) | 5 | 11.1 | 20 |
| $V_{CE0}$ (V) | 0.9 | 2.3 | 0 |
| $V_{F0}$ (V) | 1.1 | 1 | 1.5 |
| $E_{onref}$ (mJ) | 37 | 11 | 1.1 |
| $E_{offref}$ (mJ) | 22 | 4 | 0.6 |
| $E_{rrref}$ (mJ) | 12 | 4 | - |
| $t_{on}$ (ns) | 295 | 75 | 51 |
| $t_{off}$ (ns) | 625 | 600 | 69 |
| $R_G$ (Ω) | 10 | 10 | 20 |

**Table 2.** Specifications of the three-phase inverter.

| Parameters | Value |
|---|---|
| DC bus voltage ($V_{dc}$) | >540 V |
| Output current of the inverter ($I_{eff}$) | 30 Arms |
| PWM frequency ($f_s$) | 10–100 kHz |
| Fundamental frequency ($f_1$) | 50–3000 Hz |
| Efficiency ($\eta$) | >90% |

The test bench consists of the 3Ph-VSIs, a TDK Lambda 600 V, 6000 W DC source, a power analyzer Tektronix *PA*4000, a dSPACE MicroLabBox to control the system, a three-phase RL load (0–50 Ω , 3 mH), and a 100 MHz Rohde & Schwarz oscilloscope. The test bench is depicted in Figure 7.

Achieving a high switching frequency (beyond 40 kHz) utilizing the Matlab/Simulink/ dSPACE RTI approach proved unattainable. Consequently, an alternative course was pursued by employing C language programming and dSPACE RTLib [24] to program the MicroLabBox. The vector control routine finds its implementation within an Interrupt Service Routine (ISR) operating at half of the switching frequency. In order to facilitate meticulous comparisons, open loop tests were conducted, entailing the application of constant reference voltages.

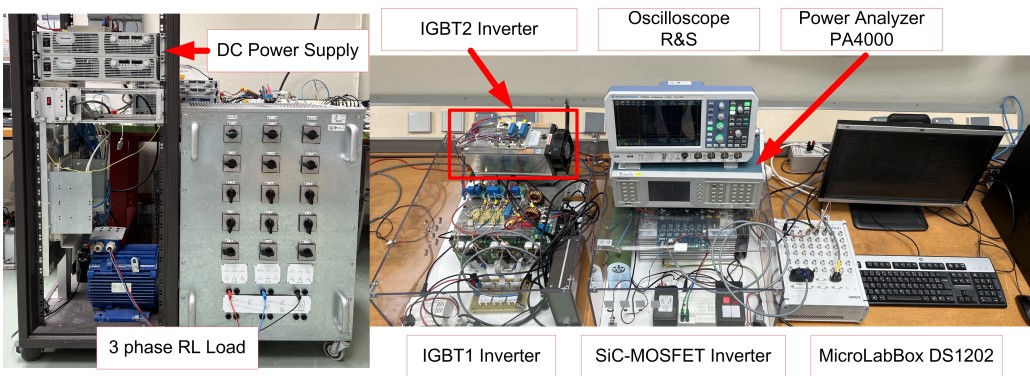

**Figure 7.** Experimental test bench.

### 4.1. Analysis of Switching Events for the SiC-MOSFET Inverter

Figure 8 displays the measurements obtained for the SiC-MOSFET inverter with a dead time ($t_d$) set at 2.5 μs during the turn-on of switch $S_{aH}$. The measurements were taken for various current values at the time of switching (positive or negative current).

Figure 8 shows the conduction of switch $S_{aH}$ and the blocking of switch $S_{aL}$. The gate control signals change, affecting the voltage of the $C_4$ ($V_{C_4}$ see Figure 2) depending on the current direction and magnitude during switching. High negative currents cause a rapid $V_{C_4}$ rise, while lower currents slow it down. A specific case matches $V_{C_4}$ rising time with the dead time. Positive currents result in a $V_{C_4}$ jump after the dead time due to switch $S_1$ activation. These results align with the theoretical study.

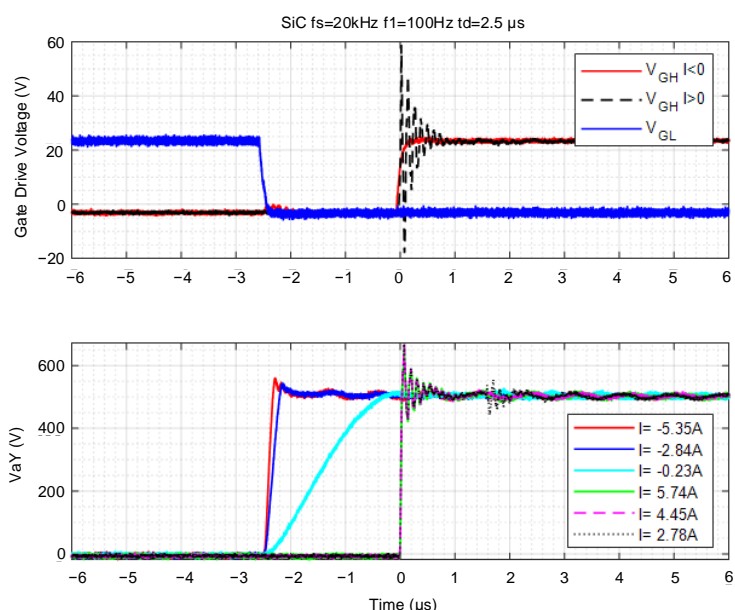

**Figure 8.** The voltage across a switch during the dead time is measured for different current levels and $t_d = 2.5$ μs in the SiC-MOSFET inverter.

The theoretical studies were validated by reducing the dead time from 2.5 μs to 1.5 μs. The results in Figure 9 show that, at low current values (cyan curves), the parasitic capacitance cannot be fully charged or discharged before the end of the dead time, resulting in a voltage $V_{C_4}$ discontinuity during switching.

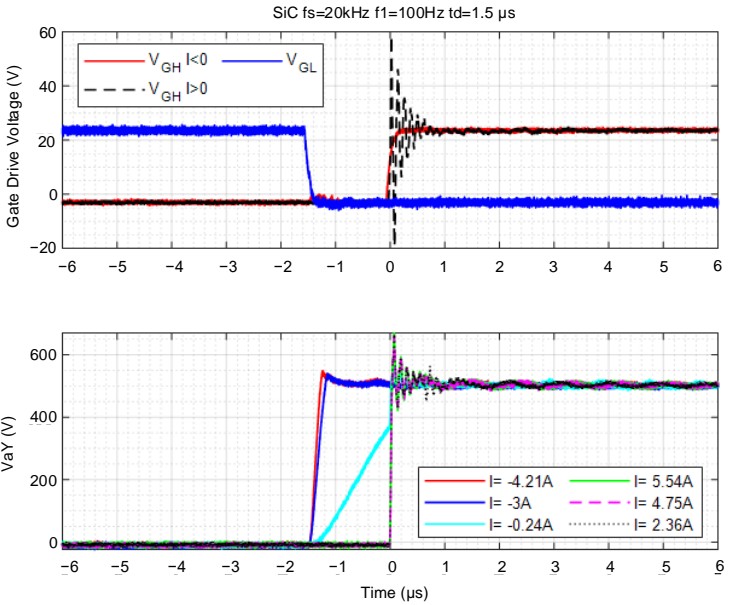

**Figure 9.** The voltage across a switch during the dead time is measured for different current levels and $t_d$ = 1.5 μs in the SiC-MOSFET inverter.

### 4.2. Measurement of the Value of Parasitic Capacitances

The relationship between $C_{out}$ (assumed $C_{out} = C_1 = C_4$), the constant current magnitude $|I_a|$ at switching, the discharge time ($\Delta t$), and the voltage variation ($\Delta V$) across the component can be expressed as follows:

$$C_{out} = \frac{\Delta t |I_a|}{2\Delta V} \tag{28}$$

The curves in Figure 10a for SiC-MOSFET, Figure 10b for IGBT1, and Figure 10c for IGBT2 were used to determine $C_{out}$ for each inverter. The discharge time, voltage variation, and current magnitude at switching were recorded from the cyan-colored curves. It was observed that the voltage variation appeared to be linear for the SiC and IGBT2 inverters, consistent with model assumptions. However, for the IGBT1 inverter, the slope increased at lower voltages. Although some studies suggest an increase in parasitic capacitance with decreasing voltage, this phenomenon will be disregarded to maintain a simple model.

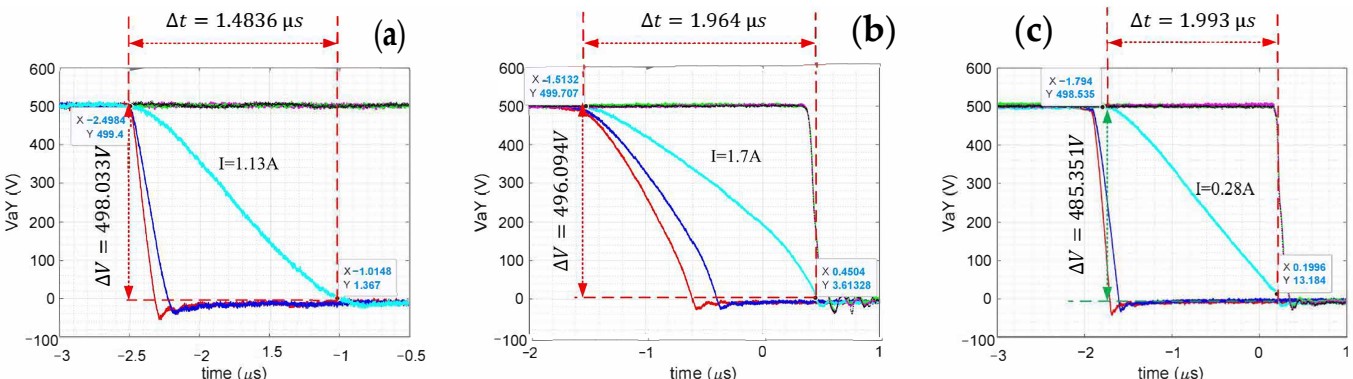

**Figure 10.** From left to right $V_{C_{out}}$ of the SiC-MOSFET inverter, the IGBT1 inverter, the IGBT2 inverter.

The impact of parasitic capacitances on the voltage drop and model accuracy was studied. Figure 11 shows the voltage drop evolution with PWM frequency for three different dead time values. The theoretical curves were plotted with and without parasitic capacitance effects and compared to PA4000 measurements.

Neglecting parasitic capacitances in the modeling leads to an evident overestimation of the voltage drop, averaging around 15%. Considering these capacitances allows for highly accurate voltage drop estimation when compared to actual measurements. Thus, ignoring this phenomenon hampers correct prediction of a three-phase inverter's output voltage value.

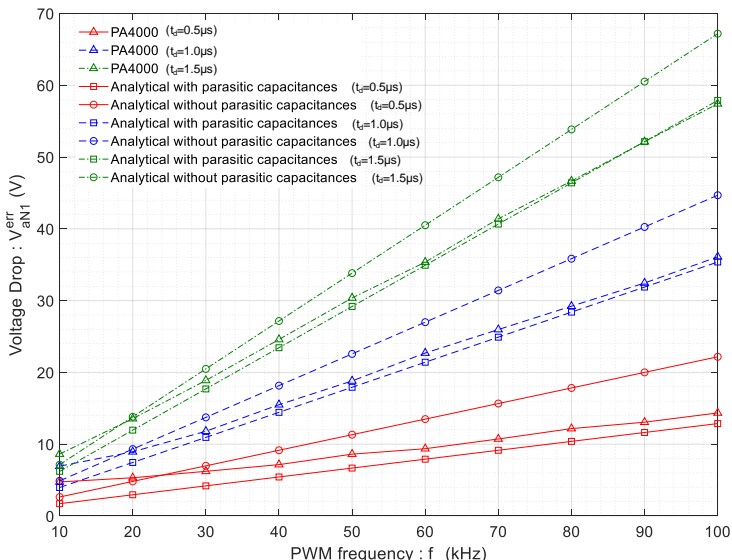

**Figure 11.** The voltage drop $V_{aN1}^{err}$ of the SiC inverter as a function of the PWM frequency for three different values of dead time.

### 4.3. Comparison of the Performances of the 3 Inverters

The SiC-MOSFET inverter exhibits the highest voltage drop, followed by the IGBT2 inverter, and then the IGBT1 inverter (Figure 12). The SiC-MOSFET inverter's voltage drop can be significantly reduced by setting the dead time to 0.5 μs, which is not feasible for IGBT1 and IGBT2 due to their substantial blocking time (around 600 ns). In conclusion, a low dead time is crucial when increasing the PWM frequency for some applications such as high-speed electric motors to avoid significant voltage drop. However, IGBT inverters are limited to approximately 1 μs of dead time due to their blocking times, which restricts their use for high-frequency drives. Increasing the DC bus voltage to compensate for the voltage drop is possible but often limited by other constraints. Control strategies that compensate for dead time effects exist, but they are beyond the scope of this thesis.

The harmonic spectra of phase voltage ($V_{aN}$) were obtained experimentally and using the analytical model concerning the dead time ($t_d$) and different modulation frequency index values ($m_f$). From these results, it can be observed that the fundamental value decreases with an increase in PWM frequency (increase in $m_f$). However, the amplitudes of the different harmonics remain nearly constant regardless of the value of $m_f$.

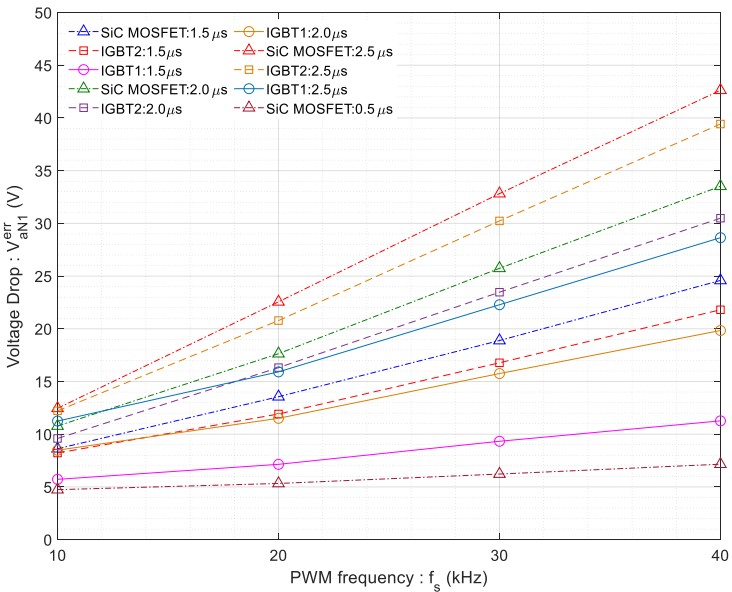

**Figure 12.** Comparison of the voltage drop for the 3 types of inverters.

### 4.4. Measurement of the Total Harmonic Distortion of Voltage for the 3 Types of Inverters

The decrease in the fundamental value with $m_f$, while the amplitudes of the different harmonics remain constant, leads to an increase in $THD_v$ with the PWM frequency for a given dead time. This observation is further supported by the $THD_v$ measurement results shown in Figure 13. For instance, when comparing the $THD_v$ value for $t_d = 2$ μs, it increases from 100% at $f_s = 10$ kHz to approximately 117% at $f_s = 40$ kHz.

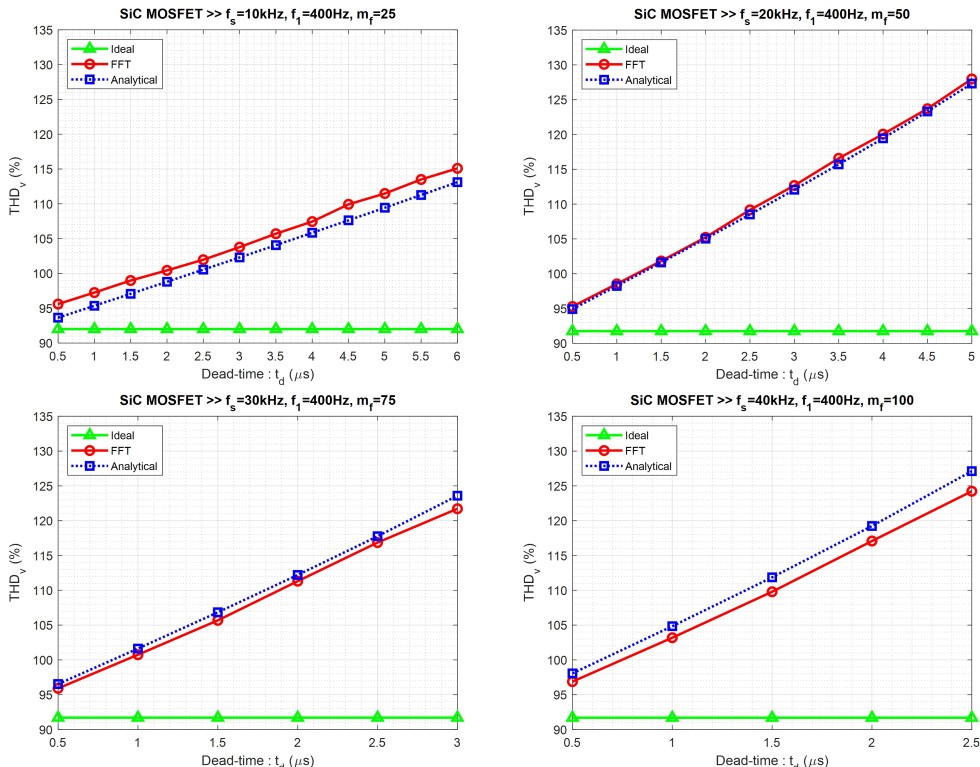

**Figure 13.** Total Harmonic Distortion of voltage (THD) for the SiC MOSFET inverter concerning dead time and at different modulation frequency indices $m_f$.

Figure 14 compares the measurement results of $THD_v$ for the three inverters under the same operating conditions. These results demonstrate that the $THD_v$ is consistently highest for the SiC-MOSFET inverter. However, selecting a very low dead time, which is possible with this type of component, can significantly improve $THD_v$. We conducted a $THD_v$ measurement for the SiC-MOSFET inverter by setting the dead time to 0.5 µs. The results are shown in Figure 14, indicating that the $THD_v$ remains at 95% regardless of the PWM frequency value.

As mentioned earlier, low-frequency voltage harmonics (5th, 7th orders, etc.) arise due to dead times. While their impact on voltage remains relatively minor in comparison to PWM-induced harmonics, they gain significance in the current waveform due to the role of the R-L load as a low-pass filter. Various dead times (ranging from 0.5 µs to 5 µs) were tested on a SiC-MOSFET inverter operating with a 400 Hz fundamental frequency and 20 kHz PWM frequency (modulation factor, $m_f = 50$). Figure 15 illustrates the results with the fundamental current amplitude normalized to 0.5 A (the current value is 4.1A). The increase in dead time amplifies the fifth- and seventh-order harmonics, overshadowing the PWM-related harmonics around the 50th order. A near-linear correlation between the amplitude of the fifth and seventh harmonics and the dead time is evident from the findings in Figure 15. The amplitude of the $n$th-order harmonic (where $n = 5, 7, 11, 13, \ldots$) for the current passing through the R-L load is governed by the following equation:

$$I_n = \frac{4}{\pi n} \cdot \frac{V_{dc}}{\sqrt{R^2 + (Ln\omega_1)^2}} \cdot \frac{t_d}{T_s} \quad ; \quad n = 5, 7, 11, 13, \ldots \quad (29)$$

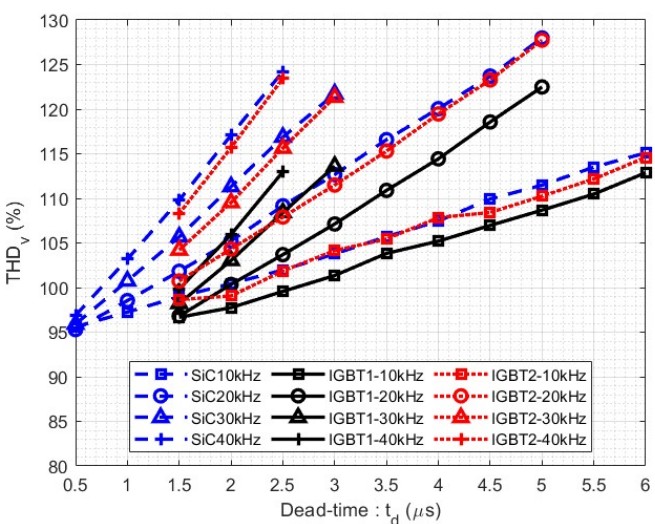

**Figure 14.** Comparison of the $THD_v$ (Total Harmonic Distortion of voltage) for the SiC-MOSFET, IGBT1, and IGBT2 inverters, varying with dead time and at different PWM frequencies.

This relationship underscores the direct proportionality between $I_n$ and the dead time value $t_d$, as corroborated by experimental observations. Applying Equation (29) using the parameter values from Table 1 and adopting $R = 27.3\ \Omega$ and $L = 3$ mH, which corresponds to the experiments in Figure 15, yields an $I_5$ value of 0.306 A for $t_d = 5$ µs. This outcome closely aligns with the experimentally recorded value of approximately 0.3 A.

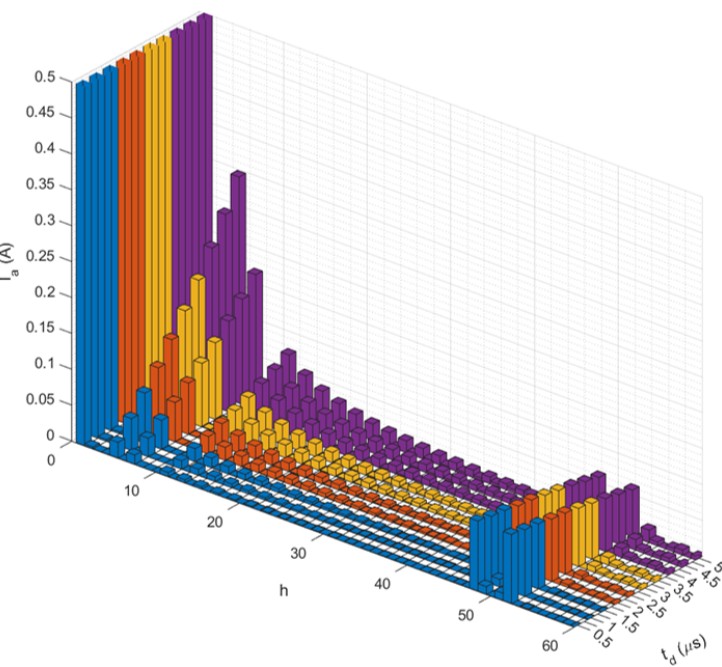

**Figure 15.** Experimental measurement of the current spectrum in the R-L load with modulation factor ($m_f = 50$) for various dead time ($t_d$) values.

## 5. Conclusions

This paper introduces an analytical model for a three-phase inverter, encompassing factors such as dead times, switching times, voltage drops across components, and parasitic capacitance. The model effectively forecasts the voltage drop at the fundamental frequency of the inverter's output voltage. Overlooking parasitic capacitance may result in an overestimation of this voltage drop. Among the contributors, dead times stand out as the primary cause of the most substantial voltage drop, influencing low-frequency harmonics that affect the load current.

The proposed model's validity is established through experimental trials conducted on three-phase inverters equipped with distinct power components (IGBT and SiC MOSFET). The results demonstrate that the SiC inverter displays a greater voltage drop compared to IGBT-based inverters under identical dead time conditions. Nonetheless, the SiC inverter can significantly diminish the dead time to 0.5 µs, leading to a reduced voltage drop. The model consistently predicts the inverter's output voltage drop and Total Harmonic Distortion (THD), with THD reduction at higher frequencies when minimizing dead time in the SiC-MOSFET inverter.

Furthermore, the impact of dead time on low-frequency harmonics is explored and verified through experimental tests, highlighting the significant influence of these harmonic amplitudes on THD. This emphasizes their role in shaping THD and underscores their importance in practical applications. In conclusion, the experimental study conducted in this paper, along with the proposed models, can serve as a valuable tool for sizing a three-phase inverter capable of powering a high-frequency electric motor. Such systems find application in actuation systems where achieving compactness through frequency enhancement and consequent weight reduction are increasingly critical factors in actuator design.

**Author Contributions:** Conceptualization N.T. and T.L.; methodology N.T. and T.L. software; P.P., L.B., T.L. and E.J.; validation, T.L. and P.P., writing—original draft preparation, E.J. and P.P.; writing—review and editing, E.J., P.P. and T.L; supervision, N.T. and T.L.; project administration, N.T. and T.L. All authors have read and agreed to the published version of the manuscript.

**Funding:** This research received no external funding.

**Conflicts of Interest:** The authors declare no conflict of interest.

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
