# Peer review of "Comparative Study of IGBT and SiC MOSFET Three-Phase Inverter: Impact of Parasitic Capacitance on the Output Voltage Distortion"

_actuators, doi:10.3390/act12090355_

Round 1
Reviewer 1 Report
(1) In equation (1), the output voltage should be phase voltages instead of pole voltage. If there are some mistakes in the notation, the eaquation should be modified.
(2) Please clarify the distinction between the pole voltage and the phase voltage in the overall explanation and equations.
(3) In line 87, I think “Rising time and falling time effect” is more appropriate expression than “Rise time and down time effect”.
(4) In equation (9), What D means? If D is the duty ratio, shouldn't the equation be modified a little because the diode doesn't continue to conduct during the off time?
(5) In equation (21), the matrix [2 –1 –1] should be replaced with 3X3 matrix.
(6) In Fig. 11 - Fig 14, the language that is not English should be translated to English.
Author Response
Dear Editor-in-Chief,
We extend our gratitude to you for permitting us to submit a revised version of our manuscript titled "Comparative Study of IGBT and SiC MOSFET Three-Phase Inverter: Impact of Parasitic Capacitance on the Output Voltage Distortion" to "MDPI-Actuators." We greatly value the time and effort that you and the reviewers have dedicated to providing valuable feedback on our manuscript.
We are particularly appreciative of the reviewers for their insightful comments, which have proven invaluable in improving our paper. In response to their suggestions, we have diligently incorporated the recommended changes into the manuscript. These modifications have been clearly marked within the document for your convenience.
Please find below a point-by-point response to the reviewers' comments and concerns.
Yours sincerely,
The Authors of the Paper
To enhance clarity, we have chosen the following color codes for corrections:
- Text in blue: Reviewer's comment
- Text in black: Author's response to comments and questions
Comments and Suggestions for Authors
- In equation (1), the output voltage should be phase voltages instead of pole voltage. If there are some mistakes in the notation, the eaquation should be modified.
First, we would like to thank the respected reviewer for his valuable comments and suggestions.
Yes, the phase voltages are included, and we have introduced a new equation, labeled as (1), to enhance clarity for our readers. Additionally, all the mistakes have been rectified.
- Please clarify the distinction between the pole voltage and the phase voltage in the overall explanation and equations.
All the mistakes have been rectified.
- In line 87, I think “Rising time and falling time effect” is more appropriate expression than “Rise time and down time effect”.
These terms are changed in the text.
- In equation (9), What D means? If D is the duty ratio, shouldn't the equation be modified a little because the diode doesn't continue to conduct during the off time?
Yes, 'D' represents the 'duty cycle.' As Fig. A1 illustrates, the diode conducts during (1-D)Ts. Therefore, the average voltage drop at the output of the inverter, attributed to the voltage drop across the components denoted as , can be expressed as follows, where 'D' signifies the duty cycle of the signal:
When > 0 ;
|
(eq1) |
when < 0 ;
|
(eq2) |
Ia>0
Ia<0
Fig. A1
- In equation (21), the matrix [2 –1 –1] should be replaced with 3X3 matrix.
The matrix is revised.
- In Fig. 11 - Fig 14, the language that is not English should be translated to English.
The figures is corrected and replaced.

Reviewer 2 Report
Dear authors, thank you for your very interesting work. I think it is very significant, well organized and well written
In order to make it ready for publication please revise a few editing details:
table 1 is partially in french, as well as it caption. It should be translated in english
text in fig 10 is too small
Author Response
Dear Editor-in-Chief,
We extend our gratitude to you for permitting us to submit a revised version of our manuscript titled "Comparative Study of IGBT and SiC MOSFET Three-Phase Inverter: Impact of Parasitic Capacitance on the Output Voltage Distortion" to "MDPI-Actuators." We greatly value the time and effort that you and the reviewers have dedicated to providing valuable feedback on our manuscript.
We are particularly appreciative of the reviewers for their insightful comments, which have proven invaluable in improving our paper. In response to their suggestions, we have diligently incorporated the recommended changes into the manuscript. These modifications have been clearly marked within the document for your convenience.
Please find below a point-by-point response to the reviewers' comments and concerns.
Yours sincerely,
The Authors of the Paper
To enhance clarity, we have chosen the following color codes for corrections:
- Text in blue: Reviewer's comment
- Text in black: Author's response to comments and questions
Comments and Suggestions for Authors
Dear authors, thank you for your very interesting work. I think it is very significant, well organized and well written
In order to make it ready for publication please revise a few editing details:
table 1 is partially in French, as well as it caption. It should be translated in English
First, we would like to thank the respected reviewer for his valuable comments and suggestions.
The text in the table and the caption has been revised.
text in fig 10 is too small
The figure is revised.

Reviewer 3 Report
The non-linearity in 3-phase systems for both SiC and Si based inverters, respectively, has been addressed in this paper.
Even though the above-mentioned topic has been profoundly discussed for recent years, it could be meaningful if authors could suggest some contribution to the actuator systems.
Thus the reviewer recommend authors to improve the introduction along with whole manuscript, in order to emphasize your contribution to actuator systems.
English in this manuscript has been written quite well. Minor modification (such as punctuation) could be updated.
Author Response
Dear Editor-in-Chief,
We extend our gratitude to you for permitting us to submit a revised version of our manuscript titled "Comparative Study of IGBT and SiC MOSFET Three-Phase Inverter: Impact of Parasitic Capacitance on the Output Voltage Distortion" to "MDPI-Actuators." We greatly value the time and effort that you and the reviewers have dedicated to providing valuable feedback on our manuscript.
We are particularly appreciative of the reviewers for their insightful comments, which have proven invaluable in improving our paper. In response to their suggestions, we have diligently incorporated the recommended changes into the manuscript. These modifications have been clearly marked within the document for your convenience.
Please find below a point-by-point response to the reviewers' comments and concerns.
Yours sincerely,
The Authors of the Paper
To enhance clarity, we have chosen the following color codes for corrections:
- Text in blue: Reviewer's comment
- Text in black: Author's response to comments and questions
- Text in red: Newly added text in the revised version
Comments and Suggestions for Authors
The non-linearity in 3-phase systems for both SiC and Si based inverters, respectively, has been addressed in this paper.
Even though the above-mentioned topic has been profoundly discussed for recent years, it could be meaningful if authors could suggest some contribution to the actuator systems.
First, we would like to thank the respected reviewer for his valuable comments and suggestions.
The introduction and conclusion of the paper have been revised as follows to address the valuable suggestions of the reviewer.
In introduction:
An expanded version of our previous conference paper [11] is presented in this paper, with the primary objective of providing a comprehensive model and investigation of the varied factors affecting voltage and current distortion in SiC-based and IGBT-based inverters designed for high-speed machinery applications [12]. These factors, including dead-time, switching delay time, voltage drop, and the parasitic capacitance of the components, are the focus of the study.
A wide range of fundamental frequencies, reaching up to $3400$ Hz, and switching frequencies ranging from $10$ kHz to $100$ kHz are covered in our experimental analysis. The operational limits of IGBT inverters when employed in high-speed motor drives, a prevalent choice in applications like electric vehicles, airplanes, electric ships, and more, are sought to be anticipated. Furthermore, the advantages of employing SiC inverters will be elucidated.
The significance of considering the parasitic capacitance of both IGBT and SiC systems is underscored to ensure precise predictions regarding inverter frequency limits. The experimental validation of these limitations, extending up to a PWM frequency of 100 kHz, represents a significant advancement in the field, building upon prior research contributions [9,13].
In conclusion:
In conclusion, the experimental study conducted in this paper, along with the proposed models, can serve as valuable tools for sizing a three-phase inverter capable of powering a high-frequency electric motor. Such systems find application in actuation systems where achieving compactness through frequency enhancement and consequent weight reduction are increasingly critical factors in actuator design.
Thus the reviewer recommend authors to improve the introduction along with whole manuscript, in order to emphasize your contribution to actuator systems.
The introduction has been revised, and three additional references have been incorporated.
